# Tamoxifen Affects Aquaporin-3 Expression and Subcellular Localization in Rat and Human Renal Collecting Ducts

**DOI:** 10.3390/cells12081140

**Published:** 2023-04-12

**Authors:** Stine Julie Tingskov, Mariagrazia D’Agostino, Frédéric H. Login, Grazia Tamma, Lene N. Nejsum, Rikke Nørregaard

**Affiliations:** 1Department of Clinical Medicine, Aarhus University, 8200 Aarhus N, Denmark; 2Department of Bioscience, Biotechnologies and Environment, University of Bari, 70125 Bari, Italy; 3Department of Renal Medicine, Aarhus University Hospital, 8200 Aarhus N, Denmark

**Keywords:** aquaporins, UUO, tamoxifen, lithium-induced NDI

## Abstract

Sex hormones play an important role in the regulation of water homeostasis, and we have previously shown that tamoxifen (TAM), a selective estrogen receptor modulator (SERM), affects the regulation of aquaporin (AQP)-2. In this study, we investigated the effect of TAM on the expression and localization of AQP3 in collecting ducts using various animal, tissue, and cell models. The impact of TAM on AQP3 regulation was studied in rats subjected to 7 days of unilateral ureteral obstruction (UUO), with the rats fed a lithium-containing diet to induce nephrogenic diabetes insipidus (NDI), as well as in human precision-cut kidney slices (PCKS). Moreover, intracellular trafficking of AQP3 after TAM treatment was investigated in Madin-Darby Canine Kidney (MDCK) cells stably expressing AQP3. In all models, the expression of AQP3 was evaluated by Western blotting, immunohistochemistry and qPCR. TAM administration attenuated UUO-induced downregulation of AQP3 and affected the localization of AQP3 in both the UUO model and the lithium-induced NDI model. In parallel, TAM also affected the expression profile of other basolateral proteins, including AQP4 and Na/K-ATPase. In addition, TGF-β and TGF-β+TAM treatment affected the localization of AQP3 in stably transfected MDCK cells, and TAM partly attenuated the reduced AQP3 expression in TGF-β exposed human tissue slices. These findings suggest that TAM attenuates the downregulation of AQP3 in a UUO model and a lithium-induced NDI model and affects the intracellular localization in the collecting ducts.

## 1. Introduction

The renal collecting duct is a critical tubular segment for water reabsorption and regulation of body water homeostasis [1]. AQP3 is localized in the basolateral membrane of principal cells in the connecting tubule and collecting duct, and together with AQP4, it represents the main exit pathway of water reabsorbed by AQP2 [2]. Several studies have shown that estrogen affects kidney-mediated water homeostasis [3,4,5], and we have previously demonstrated that estrogen affects the expression and trafficking of AQP2 in collecting-duct principal cells in rats subjected to ovariectomy [5]. In line with this, we have recently shown that tamoxifen (TAM), a selective estrogen receptor modulator (SERM), could rescue AQP2 expression and improve urinary concentration capacity in rats with lithium-induced nephrogenic diabetes insipidus (NDI) [6] as well as in rats subjected to unilateral ureteral obstruction (UUO) [7].

In addition, it has been reported that estrogen can attenuate the downregulation of AQP3 in the bladders of rats subjected to ovariectomy [8]. Moreover, it has been demonstrated that estradiol increases AQP3 mRNA and protein expression in cells from human endometrial adenocarcinoma [9] as well as in estrogen receptor (ER)-positive breast cancer cells [10]. Additionally, Huang et al. identified an estrogen response element (ERE) in the promoter region of the AQP3 gene that mediates estrogen-induced cell migration and invasion in ER-positive breast cancer cells [10]. Together, these studies indicate that estrogen can affect AQP3 regulation. It is noteworthy that TAM is one of the most used adjuvant treatments for ER-positive breast cancer [11], but the question remains whether TAM plays a role for the regulation of AQP3. 

Based on these observations, we hypothesize that TAM can rescue AQP3 expression in pathological conditions. Therefore, we set out to examine the effect of TAM on the expression and localization of AQP3 in collecting ducts using two different kidney disease models, namely the UUO model and the lithium-induced NDI model, which are associated with reduced AQP3 expression [12,13]. Furthermore, intracellular trafficking of AQP3 after TAM treatment was investigated in Madin-Darby Canine Kidney (MDCK) cells stably expressing AQP3 tagged with enhanced green fluorescent proteins (EGFP). Lastly, the impact of TAM on AQP3 regulation was examined in an ex vivo model of human renal fibrosis, namely human precision-cut kidney slices (PCKS).

## 2. Materials and Methods

### 2.1. Animals—Experimental Design and Surgical Procedures

All procedures were performed in accordance with the Danish animal welfare act, and carried out in cooperation with a veterinarian. Experimental protocols were approved by both the local and national ethical review committee (Danish Animal Experiments Inspectorate) and conducted under the authority of the Project License (No. 2015-15-0201-00658).

For the UUO study, adult male rats were used. Their starting weight was 180.9 ± 1.42 g. The rats had free access to a standard rodent diet (Altromin, Lage, Germany) and tap water. During the experiments, they were housed in cages in groups of two, with a 12 h:12 h light-dark cycle, a temperature of 21 ± 2 °C, and a humidity of 55 ± 2%. The rats were divided into four experimental groups: Sham and UUO with vehicle treatment, and sham and UUO with TAM treatment. Tamoxifen (Sigma Chemical Co. T5648, St. Louis, MO, USA) was dissolved in ethanol (EtOH) and mixed with corn oil (Merck KGaA, Darmstadt, Germany). TAM (50 mg/kg) was administered daily via oral gavage. Treatment was initiated 5 days prior to UUO surgery and continued 7 days after UUO. Control groups received vehicle (ethanol in corn oil). On the day of surgery, rats were anesthetized with 3.5% sevoflurane (Sevorane, AbbVie, Copenhagen, Denmark) mixed with atmospheric air at 2 L/min, and injected s.c. with 50 µg/kg buprenorphine (Temgesic, Indivior UK Limited, Berkshire, UK) in order to minimize post-operative pain. An abdominal incision was made to expose the left ureter, and a silk ligature was tied around the ureter to induce obstruction. A sham operation was performed using the same method without ligation of the ureter. After 7 days, the rats were sacrificed by cervical dislocation and the kidneys were removed and prepared for Western blotting and immunohistochemistry. The same cohort of rats was used in our published work investigating the effect of TAM on AQP2 regulation and fibrosis progression [7,14].

For the NDI study, adult male rats with an initial weight of 198 ± 11.5 g were used. To induce NDI, lithium chloride (LiCl; Sigma-Aldrich, Copenhagen, Denmark) was added to the chow with a final concentration of 40 mmol/kg dry food, which was provided to the rats for 14 days. Control rats were fed standard chow (Altromin, Lage, Germany) for 14 days. After 7 days, the LiCl-fed rats were divided into two groups and treated with either 25 mg/kg or 50 mg/kg TAM for the last 7 days. Tamoxifen was administrated as described above. After 14 days, the rats were sacrificed, and the kidneys were collected and prepared for Western blotting and immunohistochemistry. The same cohort of rats was used in our published work investigating the impact of TAM on AQP2 regulation and natriuresis in NDI rats [6,15].

### 2.2. Protein Isolation and Semiquantitative Immunoblotting

Renal inner medulla tissue was homogenized in dissecting buffer (0.3 M sucrose, 25 mM imidazole, 1 mM EDTA, pH 7.2) containing protease inhibitors: phosphatase inhibitor cocktails 2 and 3 (Sigma Aldrich, St. Louis, MO, USA) and complete mini protease inhibitor cocktail tablets (serine, cysteine and metalloprotease inhibitor, Roche, Hvidovre, Denmark), using a TissueLyser LT (Qiagen, Hilden, Germany). Afterwards, samples were centrifuged and 2% SDS sample buffer was added to the supernatant. The protein concentration of the homogenate was measured using a Pierce BCA protein assay kit (Roche). Total protein was separated by SDS/PAGE using 12% Criterion TGX stain-free gels (Bio-Rad Laboratories, Copenhagen, Denmark) and subsequently blotted onto a nitrocellulose membrane (Hybond ECL, GE Healthcare, Hatfield, UK). Afterwards, the blots were blocked with nonfat dry milk in PBS-T (80 mM Na_2_HPO_4_, 20 mM NaH_2_PO_4_, 100 mM NaCl, 0.1 Tween 20, adjusted to pH 7.4). The blots were then incubated with primary antibodies against AQP3 (1591AP) [16] overnight at 4 °C. Subsequently, secondary horseradish peroxidase (HRP)-conjugated antibodies were added, and the antigen-antibody complex was visualized using an enhanced chemiluminescence system (ECLPlus, GE Healthcare). All Western blots were normalized to total protein, as measured using stain-free technology [17].

### 2.3. Immunohistochemistry and Immunofluorescence on Kidney Tissue

For immunolabeling, the kidneys were fixed by retrograde perfusion via the abdominal aorta using 4% paraformaldehyde (PFA) in 0.01 M PBS buffer, washed in PBS, dehydrated using a series of graded alcohol, and embedded in paraffin. Endogenous peroxidase activity was blocked using 35% H_2_O_2_ dissolved in methanol. Afterwards, the sections were boiled in TEG buffer (1 mmol/L Tris, 0.5 mmol/L ethylene glycol tetraacetic acid, pH 9.0) for 10 min for antigen retrieval. Nonspecific binding of immunoglobulin was prevented by incubating the sections with 50 mmol/L NH_4_Cl for 30 min followed by blocking in PBS supplemented with 1% bovine serum albumin (BSA), 0.2% gelatin, and 0.05% saponin. Sections were incubated with primary antibodies against AQP3 (AQP3-003, Alomone Labs, Jerusalem, Israel) and AQP4 (AQP4-004, Alomone Labs, Jerusalem, Israel) followed by incubation with an HRP-conjugated secondary antibody. After rinsing, the sections were incubated with 3,3′-diaminobenzidine for 10 min in order to visualize bound HRP, and counterstained with Mayer’s hematoxylin. Light microscopy was performed using an Olympus BX50 light microscope and CellSens imaging software.

For immunofluorescence labeling, sections were incubated with a primary antibody against AQP3 (AQP3-003, Alomone Labs, Jerusalem, Israel), co-stained with Na/K-ATPase (05-369, Millipore), and then incubated with goat anti-rabbit IgG Alexa 488-conjugated secondary antibody (Thermo Fisher Scientific, Cambridge, MA, USA). Then, sections were counterstained with DAPI in order to visualize the nucleus.

### 2.4. Cell Culture

MDCK cells stably expressing AQP3-EGFP or Lyn-EGFP were previously described [18]. Lyn-EGFP-MDCK cell line was used as a control; the Lyn kinase membrane anchorage sequence promoted EGFP association to the plasma membrane. MDCK cells were cultured in DMEM with 1 g/L glucose (Thermo Fisher Scientific) supplemented with 10% fetal bovine serum (FBS, Thermo Fisher Scientific) and a cocktail of 0.5 U/mL penicillin (Merck), 0.5 g/mL streptomycin (Thermo Fisher Scientific). Cells were grown with 5% CO_2_ at 37 °C. Cells were allowed to grow to a maximum of 80% confluency and passaged. Cells were seeded onto collagen-coated 10 mm diameter glass coverslips placed in 24-well plates. Upon seeding, cells were either treated with 10 ng/mL of TGF-β or remained untreated. Twenty-four hours post seeding, 1 µM of TAM was added in the growth medium of a subset of untreated and TGF-β-treated cells. Twenty-four hours post seeding, a subset of untreated cells was incubated with medium containing either 20 mM of LiCl alone or in combination with TAM (1 µM). Forty-eight hours post seeding, all cells were fixed in 4% PFA for 20 min. Untreated cells were used as negative control.

To characterize to which intracellular compartment AQP3-EGFP localized, plasmids encoding endosome and lysosome markers were transfected into the AQP3-EGFP-MDCK and Lyn-EGFP-MDCK cell lines. Cells were seeded as described above and transfected using 1 µg of pmCherry-N1-Rab5 (early endosome marker), pmCherry-N1-Rab7 (late endosome marker), pmCherry-N1-Rab11 (recycling endosome marker), or pmCherry-N2-Lamp1 (lysosome marker; all from Addgene, Teddington, UK) in combination with Lipofectamine 2000 (Thermo Fisher Scientific). Lamp1 cDNA was amplified by PCR using pEGFP-N2-Lamp1 plasmid (#34831, Addgene; [19] as template and the following primers: for-5′ CCGCTCGAGATGGCGGCCCCCGG-CAGCG and rev-5′ CGCGGATCCCGATAGTCTGGTAGCCTGCGTGACTCCTCTTCCTGC. Lamp1 was inserted into pmCherry-N2 plasmid (Clontech) after digestion with BamHI/XhoI restriction enzymes (Thermo Fisher Scientific) generating Lamp1-mCherry. Lamp1-mCherry sequence was verified by sequencing (Eurofins Genomics, Constance, Germany). Competent cells of the Escherichia coli NM522 strain (New England Biolabs) were used to amplify the different plasmids. Forty-eight hours post transfection, cells were fixed with 4% paraformaldehyde for 20 min.

### 2.5. Microscopy

Labelled tissue sections were observed on an Olympus BX61 microscope (Olympus, Tokyo, Japan), and image processing was performed using Xcellence Rt software version 1.18 (Olympus).

Fixed cells were permeabilized with PBS containing 0.1% Triton X-100 and 3% BSA for 10 min. Cells were subsequently stained with 2 mg/mL Hoechst (Thermo Fisher Scientific) to visualize nuclei. To visualize the actin cytoskeleton, some samples were also stained with phalloidin-rhodamine (Merck). After labeling, samples were mounted on glass slides using Glycergel Mounting Medium (Agilent Technologies, Santa Clara, CA, USA). Stained cells were imaged with a Nikon Eclipse Ti-E system equipped with a 100× oil immersion objective and a Zyla sCMOS camera (Oxford Instruments, Abingdon, UK), controlled by the NIS-Elements software from Nikon. The fluorescence illumination system was a CoolLED pE-300white (CoolLED, Andover, UK). Fluorescence filter sets for DAPI, GFP/FITC, and TexasRed were used to detect Hoechst, EGFP, and rhodamine, respectively. All images contained 9 z-stacks with a 500 nm step-size in z.

### 2.6. Image Analysis

Images were analyzed using ImageJ software (National Institutes of Health, Bethesda, MD, USA) utilizing the Fiji image processing package [20]. Quantification of AQP3-EGFP intensity in the cell–cell junction was performed on maximum Z-projected images. Three line scans were made across one cell–cell junction (for 10 junctions for each image), and maximum intensities from the three line scans were averaged. Quantification of AQP3-EGFP intensity in the cytoplasm was similarly carried out by drawing three squares within the cytoplasm (for 10 cells for each image). These measurements were made from 4 independent experiments and 6 images for each treatment per experiment.

### 2.7. Precision-Cut Kidney Slices

PCKS were prepared from functional (eGFR > 60 mL/min/1.73 m^2^) and macroscopically healthy renal cortical tissue obtained from both female and male patients following tumor nephrectomies, as previously described [21,22]. In short, PCKS were obtained using a 6 mm biopsy punch, and sliced in ice-cold Krebs–Henseleit buffer (25 mM D-glucose, 25 mM NaHCO_3_, 10 mM HEPES, saturated with 95% O_2_ and 5% CO_2_) using the Krumdieck Tissue slicer. Slices were cultured in William’s medium E containing GlutaMAX, 10 µg/mL ciprofloxacin and 2.7 g/L D-(+)-Glucose in an 80% O_2_, 5% CO_2_ atmosphere at 37 °C. The slices were kept in constant moderate motion and the medium was replaced every 24 h. The use of human tissue for the preparation of PCKS was approved by the Central Denmark Region Committees on Biomedical Research Ethics and The Danish Data Protection Agency (Journal number 1-10-72-211-17). Informed consent was obtained from all participants involved in the study. These patients as well as demographical data were included in our previous study [14].

### 2.8. Quantitative PCR

Total RNA was harvested from human cortical tissue using a Nucleospin RNA II mini kit following the manufacturer’s protocol (Macherey Nagel, Düren, Germany). The concentration of RNA was quantified by spectrophotometry at 260 nm. cDNA synthesis was performed using 0.5 µg RNA with the AffinityScrips qPCR cDNA synthesis kit (Life Technologies, Thermo Fisher Scientific, Cambridge, MA, USA). For the qPCR reaction, 100 ng cDNA was used in combination with SYBR Green qPCR Master Mix according to the manufacturer’s instructions (Life Technologies). The reaction was run on an Aria Mx3000P qPCR System (Agilent Technologies, Santa Clara, CA, USA). AQP3 mRNA expression was calculated relative to RPL22, which was used as reference gene. The primer sequences used were: AQP3, for-5′ ACTCCAGTGTGGAGGTGGAC and rev-5′ AGTGACAG-CAAAGCCAAAGG; RPL22, for-5′ GGAGCAAGAGCAAGATCACC and rev-5′ TGTTAG-CAACTACGCGCAAC.

### 2.9. Statistical Analysis

Values are presented as means ± standard error of mean (SEM). Multiple comparisons between the in vivo experimental groups were performed using one-way ANOVA followed by a Tukey’s multiple comparisons test (LiCl study), or two-way ANOVA followed by a Tukey’s multiple comparisons test (UUO study). Additionally, data obtained using human PCKS were compared using paired *t*-test. Data obtained using MDCK cells were compared using one-way ANOVA followed by Dunnett’s multiple comparisons test. GraphPad Prism software (version 6.01) (GraphPad Software, La Jolla, CA, USA) was used for all statistical analyses. *p* values < 0.05 were considered significant.

## 3. Results

### 3.1. Tamoxifen Rescues AQP3 Expression and Affects Subcellular Localization in Both UUO and Lithium-Induced NDI Rats

To investigate the effect of TAM on the expression of AQP3 in inner medullary collecting ducts in UUO rats as well as in LiCl-treated rats, we measured the protein level using Western blotting. As shown in Figure 1A,B, AQP3 levels were significantly decreased in both UUO and LiCl-treated rats, consistent with previous studies [12,13]. Treatment with TAM partially restored AQP3 protein levels in the UUO rats but had no effect in the LiCl-treated rats (Figure 1A,B).

Next, we investigated whether TAM affects the localization of AQP3 using immunohistochemical analyses. Immunoperoxidase staining demonstrated weak labeling of AQP3 in the basolateral membrane of the collecting duct principal cells in both UUO and LiCl-treated rats compared to the sham/control rats (Figure 1D,G vs. Figure 1C,F). After TAM treatment, AQP3 labeling intensity appeared stronger and the localization of AQP3 seemed to be more lateral and apical in some cells compared to vehicle-treated UUO rats and rats treated with LiCl alone, where AQP3 mainly localized basolaterally (Figure 1E,H,I vs. Figure 1D,G). We did not observe any change in the localization between the two different doses of TAM (25 and 50 mg/kg) in the LiCl-treated rats. These data suggest that TAM partially restored AQP3 protein levels in the UUO model, and affects the localization of AQP3 in both UUO and lithium-induced NDI rats.

### 3.2. Tamoxifen Affects the Expression and Localization of AQP4 and Na/K-ATPase in Both UUO and Lithium-Induced NDI Rats

In order to investigate whether TAM also affects the localization of other basolateral proteins in the renal collecting duct, we evaluated the localization of AQP4 and the Na/K-ATPase in UUO and LiCl-treated rats. Immunoperoxidase staining showed that the AQP4 staining was decreased in the basolateral membrane of the collecting duct cells in both UUO and LiCl-treated rats compared to sham/control rats (Figure 2B,E vs. Figure 2A,D). After TAM treatment, the levels of AQP4 were more pronounced and seemed to be more laterally localized compared to vehicle-treated UUO rats (Figure 2B,C). However, the labeling intensity was lower than sham. A similar staining pattern was observed in LiCl-fed rats treated with TAM at the highest dose (50 mg/kg; Figure 2G).

Next, we investigated the localization of the Na/K-ATPase in combination with AQP3. The Na/K-ATPase is normally restricted to the basolateral plasma membrane in the majority of epithelial cells [23]. Immunofluorescence microscopy demonstrated that the Na/K-ATPase (green), as AQP3 (red), is localized in the basolateral membrane in sham (Figure 3A–C) and control rats (Figure 3J–L). After subjecting rats to UUO or LiCl, the levels of both proteins were markedly lower in the membrane (Figure 3D–F,M–O). TAM treatment, however, partially restored levels of both AQP3 and Na/K-ATPase. Moreover, TAM altered the localization of both proteins resulting in a more lateral and in some cells, also an apical distribution in the collecting ducts (Figure 3G–I,P,R). Taken together, these results indicate that TAM can rescue the expression of basolateral proteins, including AQP3, AQP4, and Na/K-ATPase in the collecting duct after UUO and LiCl treatment. Furthermore, TAM treatment seems to result in a redistribution of these proteins, so that, in addition to the basal localization, they also localize to the lateral, and, in some cells, even the apical plasma membrane.

### 3.3. AQP3 Trafficking and Subcellular Localization Are Not Affected by Tamoxifen in Transfected MDCK Cells

Next, we set out to explore if TGF-β and LiCl in combination with TAM affects the subcellular localization of AQP3, using MDCK cells stably overexpressing AQP3-EGFP or Lyn-EGFP (hereafter referred to as AQP3 cells and LynEGFP cells, respectively). Notably, these cells can only be used to study protein localization and stability, and not effects on gene expression, since expression is under the control of the cytomegalovirus (CMV) promoter. The Lyn-EGFP-MDCK cells, in which EGFP is targeted to the plasma membrane via the anchoring sequence of the Lyn tyrosine-protein kinase, was used as control [18]. Both cell lines were treated with TGF-β to mimic fibrosis induced in the UUO model or with LiCl (20 mM). Phalloidin was used to visualize the actin cytoskeleton, which is involved in the regulation of the cell structure and trafficking networks [24]. As shown in Figure 4A–F, TGF-β did not impact LynEGFP levels and localization. In contrast, a significant increase in the EGFP signal was detected in both the plasma membrane and the cytoplasm of AQP3-EGFP cells treated with TGF-β (Figure 4G–I). Interestingly, the addition of TAM to the TGF-β treatment did not attenuate the effect of TGF-β (Figure 4J,M,N). Moreover, the TGF-β treatment promoted the formation of cytoplasmic AQP3-EGFP puncta, which was even more pronounced in TGF-β+TAM-treated cells (Figure 4I,J). No significant difference in AQP3-EGFP levels and localization was detected following LiCl and LiCl+TAM treatment (Figure 4K,M,N). Taken together, these results indicate that TGF-β and TGF-β+TAM enhanced AQP3-EGFP stability, leading to more AQP3-EGFP in the membrane and the cytoplasm, which was not the case for LiCl treatment.

We also attempted to characterize the cytoplasmic structures in which AQP3-EGFP accumulated following TGF-β and TGF-β+TAM treatment. As shown in Figure 5, cytoplasmic AQP3 did not co-localize with any of the tested endosome and lysosome markers, suggesting that AQP3 resides in other structures (Figure 5).

### 3.4. Tamoxifen Rescues AQP3 Expression in TGF-β-Treated Human PCKS

Finally, we investigated whether the effect of TAM on AQP3 regulation could also be observed in human PCKS. As shown in Figure 6, treatment with 10 ng/mL TGF-β for 48 h reduced the mRNA expression of AQP3 compared with untreated kidney slices. TAM treatment appeared to rescue AQP3 mRNA expression; however, the effect did not reach statistical significance due to one patient not responding to the treatment (Figure 6A). Fluorescence microscopy revealed that TGF-β treatment reduced the protein expression of AQP3 compared to control (Figure 6B,C), which could be reversed by TAM treatment (Figure 6C,D). Our data showed no difference between men and women. These data suggest that TAM treatment rescues AQP3 expression in TGF-β-exposed human PCKS.

## 4. Discussion

Body water homeostasis is severely impacted in various renal diseases. Interestingly, sex hormones play an important role in orchestrating channel-mediated water reabsorption in the kidneys; therefore, they might be used to restore water homeostasis during disease [25]. To investigate this hypothesis, we studied the effect of TAM on AQP3 expression and localization in renal collecting ducts under pathophysiological conditions.

Our results demonstrated that TAM treatment could rescue AQP3 expression in different models of renal fibrosis (e.g., rats subjected to UUO and TGF-β-exposed human PCKS) as well as in rats with LiCl-induced NDI. These findings are in line with our previous work, showing that TAM has a positive effect on the urinary concentration capacity by increasing the level of AQP2 in both the UUO model and in a LiCl-induced NDI model [6,7]. Importantly, our previous data demonstrated that TAM increased the trans-epithelial water transport and was able to rescue the adverse effects of lithium-induced polyuria after the establishment of LiCl-induced NDI, indicating that TAM could be a useful therapeutic approach for patients with lithium-induced NDI [6].

Under normal conditions, AQP3 is located at the basolateral plasma membrane of collecting duct principal cells [26]. Here, we demonstrated that TAM not only influenced the basolateral localization of AQP3 but also other basolateral proteins, such as AQP4 and Na-K-ATPase. Following TAM treatment, all these proteins appeared to be more abundant in the kidneys of both rats subject to UUO and rats with LiCl-induced NDI. This suggests that TAM not only affects AQPs specifically, but exerts a general effect on basolateral proteins in the collecting duct in different renal disease models, perhaps via a general effect on cellular polarity or protein trafficking.

The mechanism by which TAM affects the protein levels of AQP3 is unknown. However, TAM exerts its effect by binding the estrogen receptors to the same binding site as estrogen [27]. Interestingly, an estrogen response element (ERE) in the AQP3 promoter has been identified in estrogen-receptor-positive breast cancer cells [10]. We did not measure AQP3 mRNA expression in our animal models, but based on the results from human tissue, a similar mechanism could underlie the regulation of AQP3 in renal collecting ducts, and one could speculate that TAM might affect AQP3 expression via the ERE. Moreover, it is not known if the effect of TAM on AQP3 is mediated through an agonistic or antagonistic stimulation of ER. However, we know from our previous publications that ERα, ERβ, and GPER are present in the collecting ducts of rats subjected to UUO and LiCl [6,7]. Besides that, it is well-reported that lithium treatment in rats can decrease the fraction of principal cells in the collecting duct and increase the fraction of intercalated cells [28]. Thus, one might speculate that TAM could influence this cell ratio, leading to an increased fraction of principal cells and thereby increased expression of AQP3.

Our in vivo data showed changes in AQP3 localization in the collecting ducts from UUO rats and rats with LiCl-induced NDI after treatment with TAM. In order to further investigate the effect of TAM on AQP3 localization, we used MDCK cells stably expressing AQP3-EGFP. After both TGF-β and TGF-β+TAM treatment, we observed AQP3 in the cytoplasmic puncta. Yet, we did not observe any kind of accumulation of AQP3 in endosomes or lysosomes in our MDCK cell model. However, if EGFP is degraded in the lysosomes we would not have been able to measure a signal. Moreover, we cannot rule out the possibility that the punctate staining might correspond to autophagosomes, as both TGF-β and TAM stimulate autophagy [29,30].

A limitation of the MDCK cell model is the fact that the AQP3-EGFP expression in the cell model is under the control of the CMV promoter; therefore, these transfected cells can only be used to study the subcellular localization of AQP3-EGFP as well as protein stability. Moreover, the cells were not polarized, which might also have affected the localization of AQP3-EGFP. These results can therefore not be compared directly with the in vivo results from the rats. In addition, one could speculate that TAM might not be active in MDCK cells, since it is a prodrug that is metabolized into more active metabolites, such as 4-hydroxytamoxifen and endoxifen in the liver [31,32]. Thus, there might be some concerns using TAM in an isolated cell system. However, some studies have shown that TAM does not need to be metabolically activated [33,34].

By using human kidney slices we showed that TAM treatment improved the labeling intensity of AQP3 in TGF-β-treated human slices, indicating that we are able to extrapolate our findings to humans. In conclusion, this study demonstrated that TAM rescued AQP3 expression in two different animal disease models as well as in human kidney slices. Moreover, TAM rescued the expression of other basolateral proteins, including AQP4 and Na/K-ATPase, after UUO and lithium treatment. Together, these results suggest that TAM might exert a general effect on basolateral proteins in the collecting duct, resulting in beneficial effects on impaired water handling.

## Figures and Tables

**Figure 1 cells-12-01140-f001:**
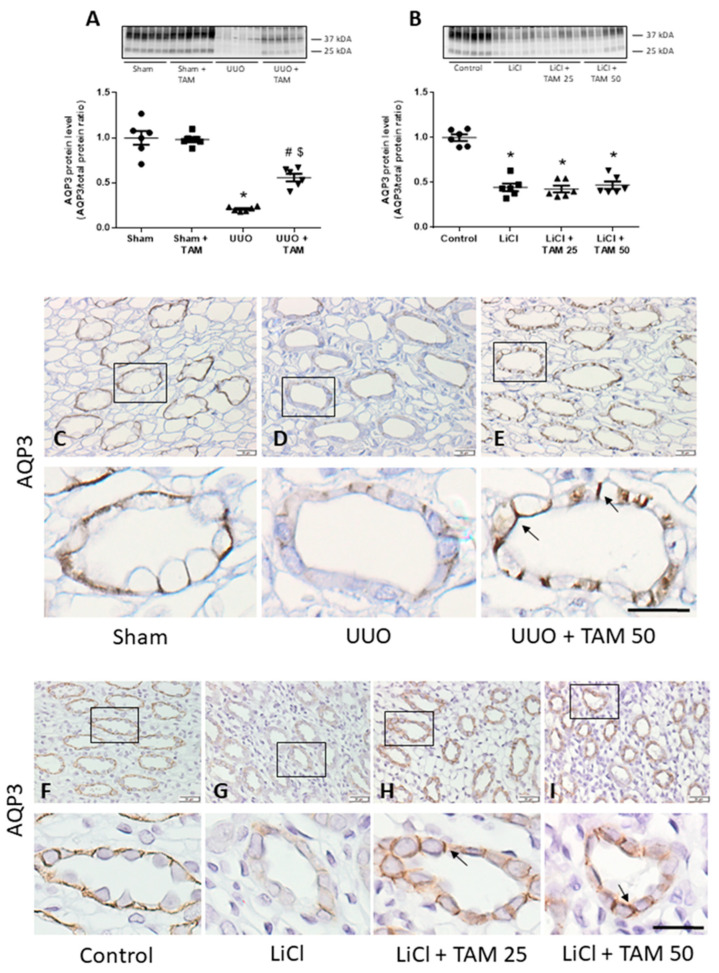
TAM rescues AQP3 expression and affects subcellular localization in both UUO and lithium-induced NDI rats. Western blot analysis revealing the impact of TAM treatment on AQP3 protein expression, relative to total protein, in (**A**) UUO rats and (**B**) LiCl-treated rats. Data are presented as mean ± SEM (n = 6 per group). * *p* < 0.05 compared to sham or control; # *p* < 0.05 compared to UUO and $ *p* < 0.05 compared to sham + TAM. (**C**–**E**) Representative immunohistochemistry images of AQP3 expression in collecting ducts in (**C**) sham rats, (**D**) UUO rats, and (**E**) UUO rats treated with TAM 50 mg/kg (n = 4 per group). (**F**–**I**) Representative immunohistochemistry images of AQP3 expression in (**F**) control rats, (**G**) LiCl-treated rats, and LiCl-treated rats treated with (**H**) TAM 25 mg/kg or (**I**) TAM 50 mg/kg (n = 4 per group). Below shows the insert. Scale bar 20 µm. Arrows indicate lateral and apical AQP3 labeling.

**Figure 2 cells-12-01140-f002:**
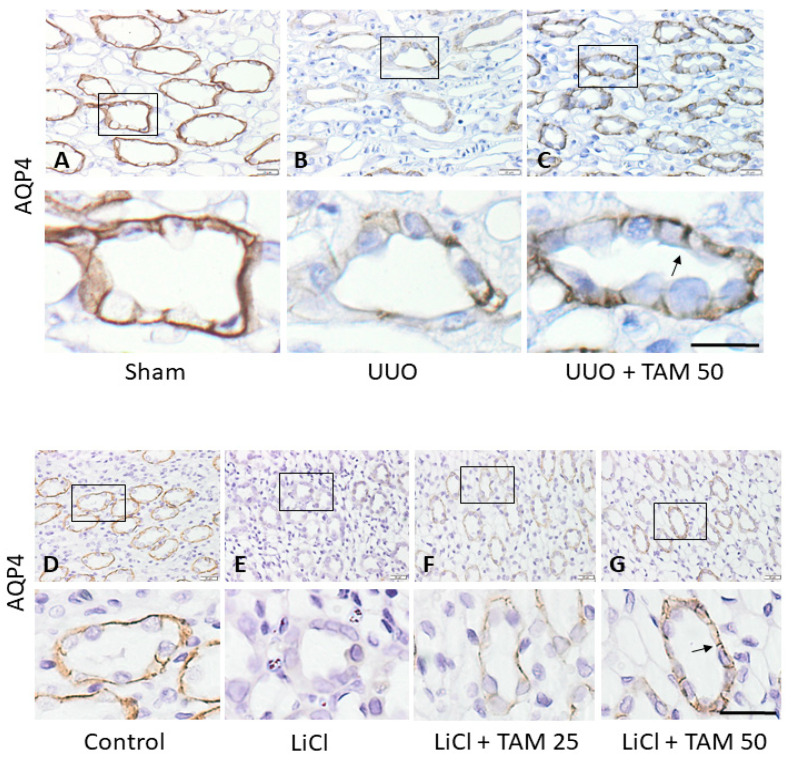
TAM affects expression and localization of AQP4 in rats subjected to UUO and LiCl treatment. (**A**–**C**): Representative immunohistochemistry images of AQP4 expression in collecting ducts in (**A**) sham rats, (**B**) UUO rats, and (**C**) UUO rats treated with TAM 50 mg/kg. Below shows the insert (n = 3–4 per group). (**D**–**G**) Representative immunohistochemistry images of AQP4 expression in (**D**) control rats, (**E**) LiCl-treated rats, and LiCl-treated rats treated with (**F**) TAM 25 mg/kg or (**G**) TAM 50 mg/kg (n = 4 per group). Below shows the insert. Scale bar 20 µm. Arrows indicate AQP4 labeling.

**Figure 3 cells-12-01140-f003:**
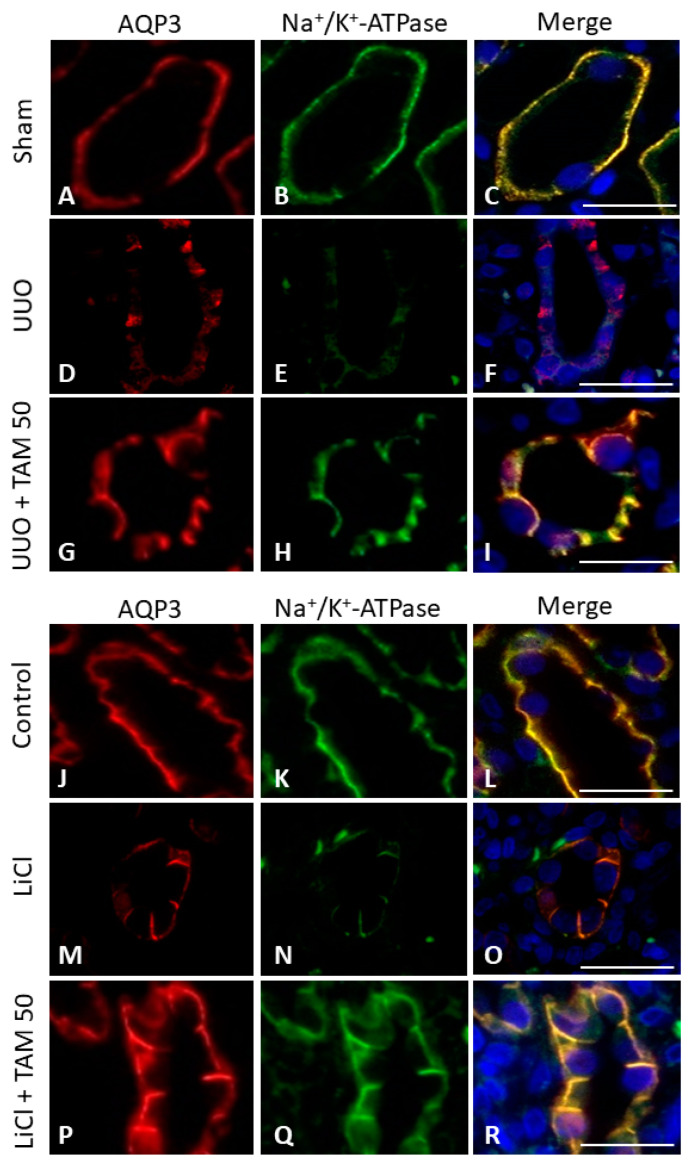
TAM affects expression and localization of AQP3 and Na/K-ATPase in rats subjected to UUO and LiCl treatment. (**A**–**I**): Representative immunofluorescence images of AQP3 (red) and Na/K-ATPase (green) expression in collecting ducts in (**A**–**C**) sham rats, (**D**–**F**) UUO rats, and (**G**–**I**) UUO rats treated with TAM 50 mg/kg (n = 3–4 per group). (**J**–**R**) Representative immunofluorescence images of AQP3 (red) and Na/K-ATPase (green) expression in collecting ducts in (**J**–**L**) control rats, (**M**–**O**) LiCl-treated rats, and (**P**–**R**) LiCl-treated rats treated with TAM 50 mg/kg (n = 4 per group). The yellow color represents co-localization between AQP3 and Na/K-ATPase. Scale bar 20 µm.

**Figure 4 cells-12-01140-f004:**
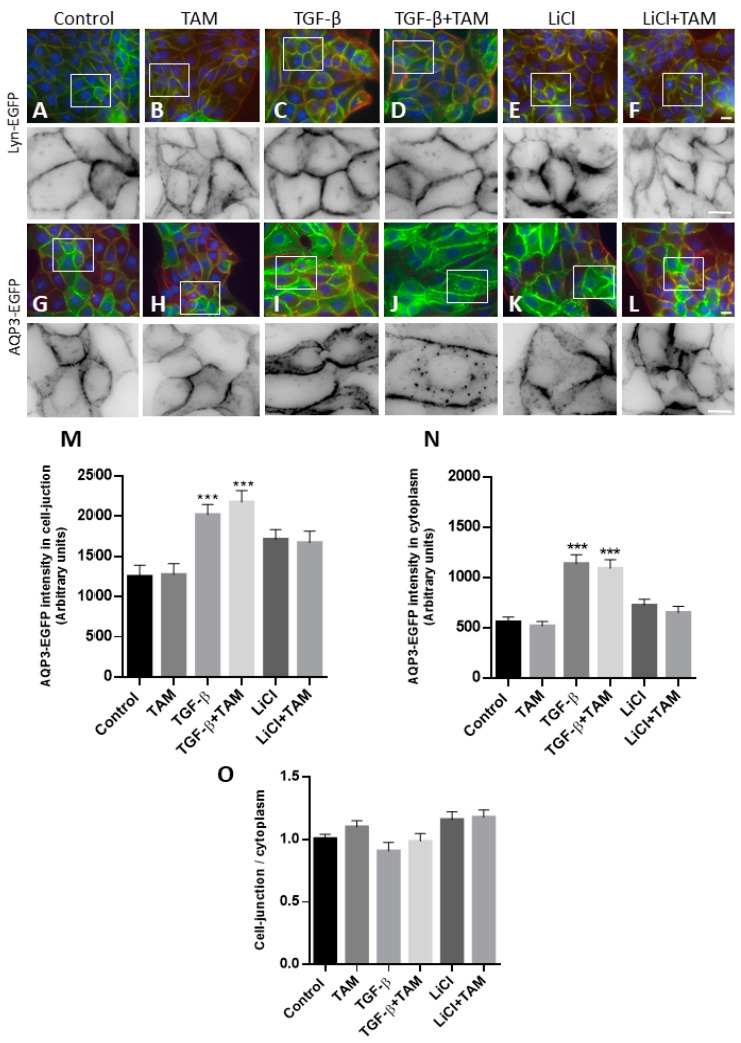
The effect of TAM treatment on AQP3-EGFP localization in stably transfected MDCK cells. (**A**–**L**): Representative immunofluorescence images of MDCK stably expressing AQP3-EGFP exposed to TGF-β, LiCl, and TAM. (**A**–**F**): Lyn-EGFP-MDCK, where EGFP is targeted to membranes by the Lyn kinase anchoring sequence. Below shows the insert in grey scale. (**G**–**L**): AQP3-EGFP in colored and in zoomed gray scale below. Green fluorescence represents AQP3-EGFP, red represents bundled actin (phalloidin), and blue color represents the nuclei (Hoechst). Scale bar 15 μm. (**M**): Quantification of AQP3-EGFP maximum intensity in the cell-cell junction of transfected cells. (**N**): Quantification of AQP3-EGFP mean intensity in the cytoplasm. (**O**): The ratio of the AQP3-EGFP intensity in the cell-cell junction over the intensity in the cytoplasm. Data are from 4 different experiments based on 6 images for each treatment per experiment. Each bar represents the mean ± SEM. *p* < 0.001 was considered statistically significant, indicated by ***.

**Figure 5 cells-12-01140-f005:**
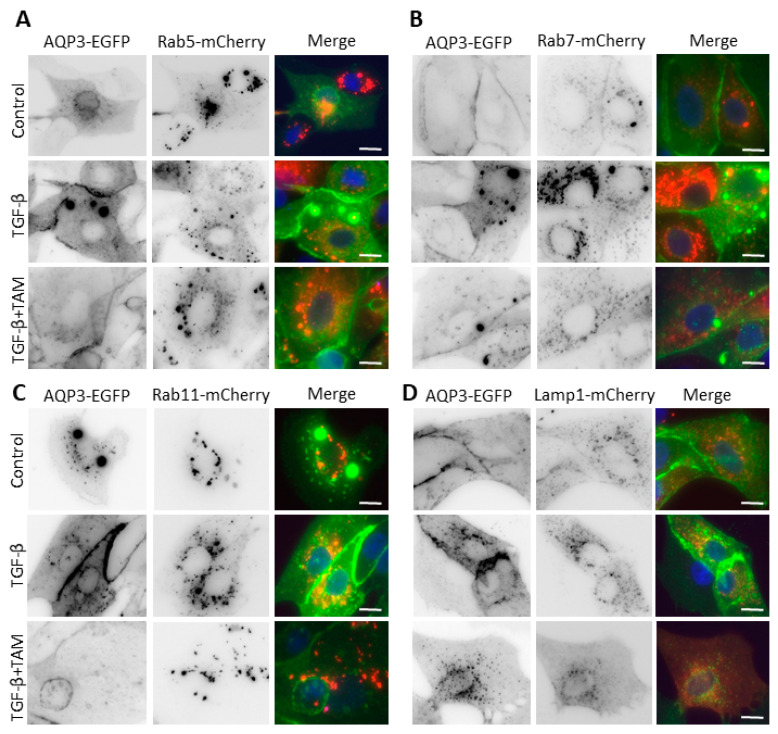
The localization of AQP3-EGFP and Rab5-mCherry, Rab7-mCherry, Rab11-mCherry, and Lamp1-mCherry in MDCK cells treated with TGF-β and TGF-β+TAM. Representative fluorescence images of AQP3-EGFP-MDCK cells transiently transfected with plasmids encoding Rab5-mCherry (**A**), Rab7-mCherry (**B**), Rab11-mCherry (**C**), and Lamp1-mCherry (**D**) exposed to TGF-β for 48 h and TGF-β+TAM after 48 h (24 h co-treatment). Forty-eight hours post transfection, cells were fixed and stained with Hoechst (blue, nuclei). Data are representative pictures from 3 different experiments. Scale bar 15 μm.

**Figure 6 cells-12-01140-f006:**
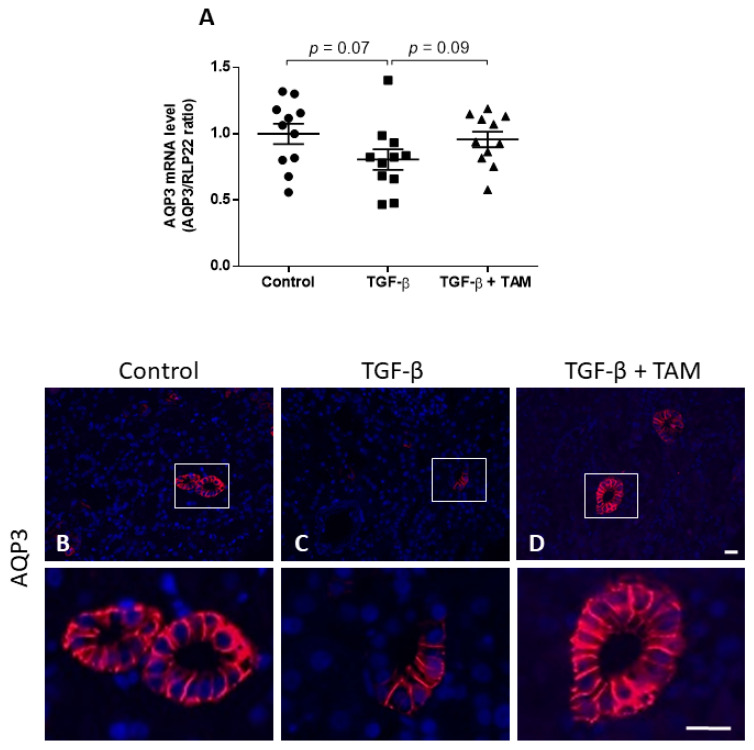
TAM rescues AQP3 expression in TGF-β-treated human PCKS. (**A**): QPCR showing AQP3 mRNA expression in human PCKS incubated with TGF-β (10 ng/mL) for 48 h in the absence and presence of TAM (5 nM) for the final 6 h. Relative expression was calculated using the reference gene RPL22. Data are presented as mean ± SEM (n = 11 patients). (**B**–**D**): Immunofluorescence images of AQP3 (red) in collecting ducts in (**B**) control PCKS after 48 h, (**C**) PCKS incubated with TGF-β for 48 h, and (**D**) PCKS incubated with TGF-β for 48 h and TAM for the final 6 h. The picture below shows the insert. Scale bar 50 μm.

## Data Availability

All data discussed are presented within the article.

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
