# Peer review of "Tamoxifen Affects Aquaporin-3 Expression and Subcellular Localization in Rat and Human Renal Collecting Ducts"

_cells, 2023, doi:10.3390/cells12081140_

Round 1
Reviewer 1 Report
General Comment: This is an interesting study of the effects of tamoxifen on AQP3. The authors also study AQP4 and Na-K-ATPase. The UUO and lithium studies use tissue from the same animals as the authors used previously to study AQP2.
Major Comments:
1. The second paragraph of the discussion briefly mentions the current findings on AQP3 in light of the authors' previous studies of AQP2. It would be useful to discuss any implications for transepithelial water transport and water handling.
2. Only male animals were used to study the effects of tamoxifen, a SERM. Is there any data to address whether the effect would be similar or different in female animals? The human samples (PCKS) did include both men and women. Were the results similar?
3. Please add the number of samples/replicates to the figure legends.
4. In figure 2, is the AQP3 staining more lateral or decreased basolateral, or both?
Minor Comment: In the introduction in paragraphs 2 and 3, several sentences start "Moreover,"
Author Response
General Comment: This is an interesting study of the effects of tamoxifen on AQP3. The authors also study AQP4 and Na-K-ATPase. The UUO and lithium studies use tissue from the same animals as the authors used previously to study AQP2.
Response: We would like to thank the reviewer for the constructive comments on our manuscript. We have answered the individual comments below.
Major Comments:
The second paragraph of the discussion briefly mentions the current findings on AQP3 in light of the authors' previous studies of AQP2. It would be useful to discuss any implications for transepithelial water transport and water handling.
Response: We thank the reviewer for the comments. We have now included a sentence about water transport and water handling in the discussion of the revised manuscript.
Only male animals were used to study the effects of tamoxifen, a SERM. Is there any data to address whether the effect would be similar or different in female animals? The human samples (PCKS) did include both men and women. Were the results similar?
Response: We thank the reviewer for this important comment. We have previously studied the effect of tamoxifen in relation to fibrosis in both female and male rats subjected to UUO. Our results showed that TAM effectively attenuates renal fibrosis in both male and female rats, indicating similar effects irrespective of gender. We have not studied the effect of TAM in relation to the regulation of AQPs in female rats. However, this will be a part of future studies.
For the human PCKS we did not observe any differences between men and women in the regulation of AQP3 and we have now included this in the revised manuscript. We have previously studied the effect of TAM in human PCKS in relation to fibrosis and here we found that TAM attenuated TGF‐β-induced fibrogenesis in human PCKS, irrespective of sex, but interindividual differences in drug efficacy were observed.
Please add the number of samples/replicates to the figure legends.
Response: Thanks for the suggestion; we have now stated all the number of replicates in all the figure legends in the revised manuscript.
In figure 2, is the AQP3 staining more lateral or decreased basolateral, or both?
Response: We thank the reviewer for the comments. After 7dUUO and LiCl the AQP4 staining is decreased basolateral. After TAM treatment the staining seemed to be more lateral compared to sham and control but the staining is more pronounced in the basolateral membrane compared to UUO and LiCl. We have clarified this in the revised version of the manuscript.
Minor Comment:
In the introduction in paragraphs 2 and 3, several sentences start "Moreover,"
Response: Thanks for the suggestion; we have now changed the word in different sentences.
Reviewer 2 Report
The study is important and provides several routes and models to test the hypothesis, that tamoxifen can rescue water transport uretral epithelium by affecting the basolateral expression of the aquaporins AQP3 and AQP4, which act in concert with AQP2 on the apical side. The investigators suggest that TAM could have affected the polarity of the cells, which, however, was not specificallly addressed in the investigation, but might be concluded from a seemingly general effect on the sorting of other primarily basolaterally localizing proteins. As a suggestion, a further study could be assessment of the integrity of tight and adherence juncions and the relative amount F-actin in relation to either of the AQPs and Na/K-ATPse, so as to address sorting and localization of AQP- containing vesicles in cells and tissues under challenge, with and without TAM treatment. Still, the the messages and the conclusions of study are clear, and very well supported by the provided evidences.
Author Response
The study is important and provides several routes and models to test the hypothesis, that tamoxifen can rescue water transport ureteral epithelium by affecting the basolateral expression of the aquaporins AQP3 and AQP4, which act in concert with AQP2 on the apical side. The investigators suggest that TAM could have affected the polarity of the cells, which, however, was not specifically addressed in the investigation, but might be concluded from a seemingly general effect on the sorting of other primarily basolaterally localizing proteins. As a suggestion, a further study could be assessment of the integrity of tight and adherence junctions and the relative amount F-actin in relation to either of the AQPs and Na/K-ATPs, so as to address sorting and localization of AQP- containing vesicles in cells and tissues under challenge, with and without TAM treatment. Still, the messages and the conclusions of study are clear, and very well supported by the provided evidences.
Response: We thank the reviewer for these nice comments and this valuable suggestion for further studies. We appreciate this suggestion and agree with the reviewer that it indeed would be very interesting to assess the integrity of tight and adherence junctions in relation to the AQPs. We will address this in future studies.